

# The biosonar of the boto: evidence of differences among species of river dolphins (*Inia* spp.) from the Amazon

Jéssica F. Melo[1,2], Thiago O. S. Amorim[1,3], Mariana Paschoalini[1,2,3] and Artur Andriolo[1,3]

[1] Laboratório de Ecologia Comportamental e Bioacústica - LABEC, Universidade Federal de Juiz de Fora, Juiz de Fora, Minas Gerais, Brazil
[2] Grupo de Pesquisa em Mamíferos Aquáticos Amazônicos, Mamirauá Sustainable Development Institute, Tefé, Amazonas, Brazil
[3] Instituto Aqualie, Juiz de Fora, Minas Gerais, Brazil

## ABSTRACT

Echolocation clicks can reflect the anatomy of the vocalizing animal, enabling the distinction of species. River dolphins from the family Iniidae are formally represented by one species and two subspecies (*Inia geoffrensis geoffrensis* and *I. g. humboldtiana*). Additionally, two other species have been proposed (*I. boliviensis* and *I. araguaiaensis*) regarding its level of restricted distribution and morph-genetics differences. For the Committee on Taxonomy of the Society for Marine Mammalogy, the specific status of the proposed species relies on further knowledge on morphology, ecology, and genetics. Given that species-specific status is required for conservation efforts, we described and compared the echolocation clicks of *Inia* spp., searching for specific differences on their vocalizations. The sounds were captured with a Cetacean Research ™ C54XRS (+3/−20 dB, −185 dB re: 1V/μPa) in Guaviare River (Orinoco basin), Madeira River (Madeira basin), Xingu River (Amazon Basin), and Araguaia River (Tocantins-Araguaia basin). We found significant differences in all analyzed parameters (peak frequency, 3 dB bandwidth, 10 dB bandwidth and inter-click interval) for all species and subspecies. Differences in acoustical parameters of clicks are mainly related to the animal's internal morphology, thus this study may potentially support with information for the species-level classification mostly of *I. araguaiaensis* (the Araguaian boto). Classifying the Araguaian boto separately from *I. geoffrensis* has important implications for the species in terms of conservation status, since it is restricted to a highly impacted river system.

## INTRODUCTION

The biosonar of odontocetes (Cetartiodactyla: Odontoceti) is a complex system for navigation and hunting. Through the analysis of echolocation clicks, it is possible to distinguish dolphin species, since the characteristics of the sound produced by the animal depend on the anatomy of its skull and organs responsible for sound production (*Lilly & Miller, 1961*; *Norris, 1968*, *1975*; *Norris et al., 1971*). The sound is reflected in different materials inside the animals' head, generating a set of several pulse paths outside the axis

Corresponding author
Jéssica F. Melo,
jessica.melo@mamiraua.org.br

with the spectral properties specific to the species (*Baumann-Pickering et al., 2010*). Internal pulse reflections can reveal the anatomy of the vocalizing animal, mainly through spectral peaks that are dependent on the morphology of the skull, and therefore show a specific aspect of the species (*Soldevilla et al., 2008*). Efforts are being made to discriminate free-range marine cetacean species through their clicks, mainly in response to the increasing use of passive acoustic monitoring (*Madsen et al., 2005*; *Zimmer et al., 2005*; *Hildebrand et al., 2015*; *Amorim et al., 2019*). Several studies have already shown that the frequency parameters of clicks are different within odontocetes. Porpoises can be distinguished at the subfamily level by peak frequency and time duration of their clicks (*Kamminga, Cohen Stuart & Silber, 1996*); *Neophocaena phocaenoides* (finless porpoise) can be distinguished from *Lipotes vexillifer* (baiji) and *Tursiops truncatus* (bottlenose dolphins) by the frequency parameters of their clicks (*Akamatsu et al., 1998*); *Phocoena phocoena* (harbor porpoise) and *Pseudorca crassidens* (false killer whale) clicks are distinguishable from four species of dolphins based on peak frequency and click duration (*Nakamura & Akamatsu, 2003*); *Grampus griseus* (Risso's dolphins) and *Lagenorhynchus obliquidens* (Pacific white-sided dolphins) can be distinguished to species level by the frequency values of the spectral peaks and notches (*Soldevilla et al., 2008*).

River dolphins are a polyphyletic group morphologically and phylogenetically distinct from marine dolphins, only found in northern South America and the subcontinent of Asia (*Hamilton et al., 2001*; *Reeves & Martin, 2009*) and its habitats are overlapped by many anthropogenic stressors (*Reeves & Leatherwood, 1994*; *Trujillo et al., 2010*). These mammals share a long and independent evolutionary history as a highly modified taxon, having more autapomorphies than shared characters useful for determining their affiliations (*Messenger, 1994*). Endemic to the Amazon, the family Iniidae are formally represented by two subspecies: *Inia geoffrensis geoffrensis* in the Amazon basin and *I. g. humboldtiana* in the Orinoco basin (both subspecies named here as Amazon river dolphin or boto). Additionally, two other species have been proposed: *I. boliviensis* (Bolivian boto) in the Madeira basin and *I. araguaiaensis* (Araguaian boto) in the Araguaia-Tocantins basin (*Pilleri & Gihr, 1977*; *Best & da Silva, 1989*; *Hrbek et al., 2014*). All lineages from the genus *Inia* are geographically separated by rapids and waterfalls among river basins, although some animals of the lineage *I. boliviensis* manage to cross the barrier that separates them from *I. g. geoffrensis* in the Madeira River, resulting in the formation of a group of hybrids biologically distinct from the species of origin (*Gravena et al., 2014*; *2015*). Such hybrid zone is also identified for the proposed species *I. araguaiaensis* and the *I. geoffrensis* in the region of the Marajó Bay–Tocantins' River mouth (J. Farias & G. Melo-Santos, 2020, personal communication).

There are few morphological differences among the proposed species—the number of teeth and the size of the rostrum are pointed out as external characteristics that distinguish them (*Pilleri & Gihr, 1977*; *Hamilton et al., 2001*; *Banguera-Hinestroza et al., 2002*; *Ruiz-García, Banguera & Cardenas, 2006*; *Hrbek et al., 2014*), and the biggest differences are molecular. Given that the morphological differences are subtle, the Araguaian and Bolivian botos are yet to be recognized by the Committee on Taxonomy of the Society for Marine Mammalogy (*Committee on Taxonomy, 2020*). The Araguaian boto is a sister
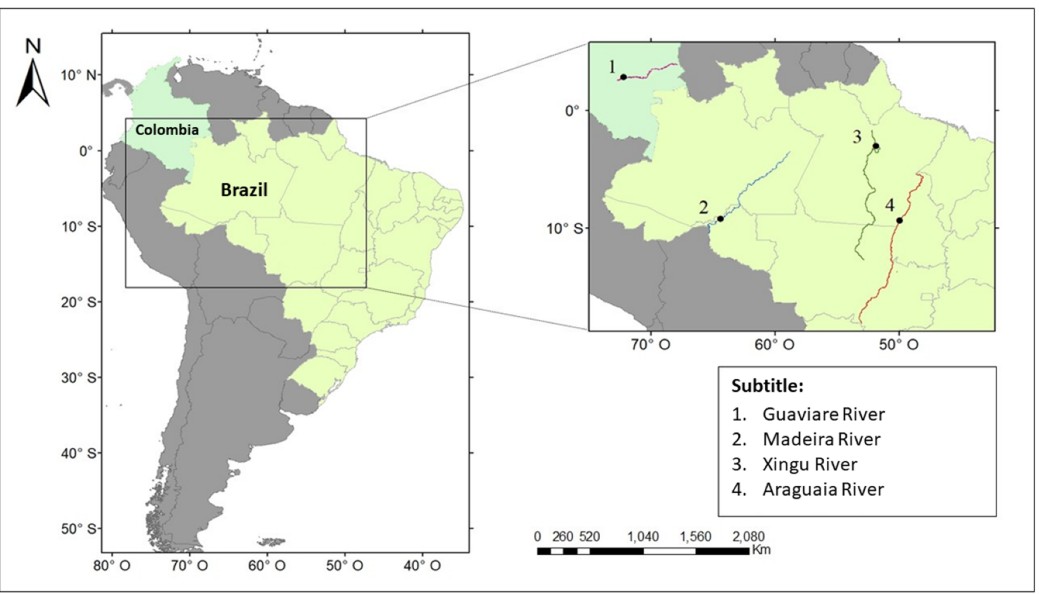

**Figure 1 Study area.** The points show where the vocalizations of *Inia* spp. were collected. The colored lines show the course of the (1) Guaviare, (2) Madeira, (3) Xingu, and (4) Araguaia rivers.

species of the boto or Amazon river dolphin and presents even more distinct characters when compared to the Bolivian boto (*Hrbek et al., 2014*).

Because of the morphological similarities among the lineages of *Inia* and the requirement of further information to assess the conservation status, acoustics analysis emerges as a complementary tool for the distinction among them that may support the evidences already published. We are not aware of studies that describe the clicks of Bolivian and the Araguaian botos. Acoustics studies with river dolphins are scarce when compared to marine dolphins. Therefore, our objective was to describe and compare the acoustical parameters of *Inia* clicks among lineages. This comparison and the description of the biosonar parameters of each lineage can improve the knowledge on their biology, in addition to being the first step towards monitoring of the species using acoustics methods.

# MATERIAL & METHODS

## Study area

Data collection was carried out in four hydrographic basins (Fig. 1). The subspecies *I. g. humboldtiana* was sampled in the Guaviare River, Orinoco basin, Colombia; *I. g. geoffrensis* in the Xingu River, Amazon basin, Brazil; *I. araguaiaensis* in the Araguaia River (Cantão State Park), Tocantins-Araguaia basin, Brazil; and *I. boliviensis* in the Madeira River, Amazon basin, Brazil. Data regarding *I. boliviensis* was sampled in the Madeira River within the artificial lake created between two hydroelectric dams: the Jirau and the Santo Antônio Dams. This human enterprise was constructed in a hybrid zone of *I. boliviensis* and *I. geoffrensis*, creating a barrier for individuals, drastically breaking the genetic flow between populations (*Gravena et al., 2014*, *2015*). Therefore, it was not

possible to obtain specific data only for *I. boliviensis*, since it is impossible to differentiate species visually. The sounds collected in the Madeira River were then attributed to *I. boliviensis, I. g. geoffrensis* and hybrids.

The Guaviare River is a sinuous white-water river of the Orinoco basin, rising in the eastern Colombian mountain range and flowing into the Orinoco River at the confluence with Inirida River. River nutrient levels are low, there is rapid flow and sandy sediments (*Medina & Silva, 1990*; *Meade & Koehnken, 1991*; *Savage & Potter, 1991*). It is 1,350 km long, with a basin area of 112,522 km$^2$, flowing at 8,200 m³/s (http://www.siatac.co/web/guest/region/hidrologia). The Xingu is a large clear-water river of the Amazon basin, covering a drainage area of approximately 520,000 km² and about 2,000 km in length with an average flow between 2,582 and 9,700 m³/s (*Pettena et al., 1980*; *Latrubesse & Sinha, 2005*). In its lower course, it presents a mosaic environment composed by rocky margins and flooded forest (várzea) (*Latrubesse & Sinha, 2005*). The Itamacará waterfall is upper the limit of the dolphins' distribution (M. Paschoalini, 2020, personal communication). The Madeira is a wide and muddy-white-water river, and one of the main tributaries in the Amazon river basin extending to three countries with 51% in Brazil, 42% in Bolivia and 7% in Peru, where the Madre de Dios River, a tributary of the Mamoré River, is born (*Guyot, 1993*). Along the Madeira River, there are 18 rapids and waterfalls that extend a distance of 290 km (*Cella-Ribeiro et al., 2013*), and most of them are currently submerged by the Santo Antônio and Jirau hydroelectric dam reservoirs (*Gravena et al., 2015*). The Araguaia River is the major tributary of the Tocantins-Araguaia basin. It is a low depth-black-water river 2,600 km in length (*Brazilian Ministry of the Environment, 2006*). In the hydrographic basin of the Araguaia River, there is a protected area of 90,000 hectares created by the Brazilian government in 1998, the Cantão State Park (*Seplan, 2001*). With approximately 880 lakes and many meanders and natural channels, the park comprises two dominant biomes, the Amazon forest in the west and the Cerrado (Brazilian savanna) in the east, bounded in the southwest by the Bananal Island region (*Seplan, 2016*).

**Data collection**

In the Guaviare River, sound samples were collected during 6 days in March 2016, at the middle reaches of this river near the Mapiripã rapids (lat 2° 52′ 54.4044″ S; long 72° 10′ 29.2656″ W). In the Madeira River data were collected along the dam's reservoirs, mainly at the Jaci-Paraná municipality (lat 9° 11′ 15.9″ S; long 9° 11′ 15.9″ S), during a 4-day effort in October 2014. In the Xingu River data were collected near the Vitória do Xingu municipality and along the river margins up to Belo Monte hydroelectric dam (lat 2° 41′ 55.824″ S; long 51° 58′ 15.1212″ W) in a 5-day effort during June 2015. Finally, the sound samples in the Araguaia River were collected inside the Cantão State Park (lat 9° 18′ 47.88″ S; long 49° 56′ 37.32″ W) in a 6-day effort in June 2017. The permit for fieldwork inside the Cantão State Park was approved by the Tocantins State Government, Instituto Natureza do Tocantins—Naturantins (permit number 1497-2017).

For data collection, a small outboard vessel was used for transportation and dolphin observations. At the presence of a group of dolphins, the engine of the vessel was turned off
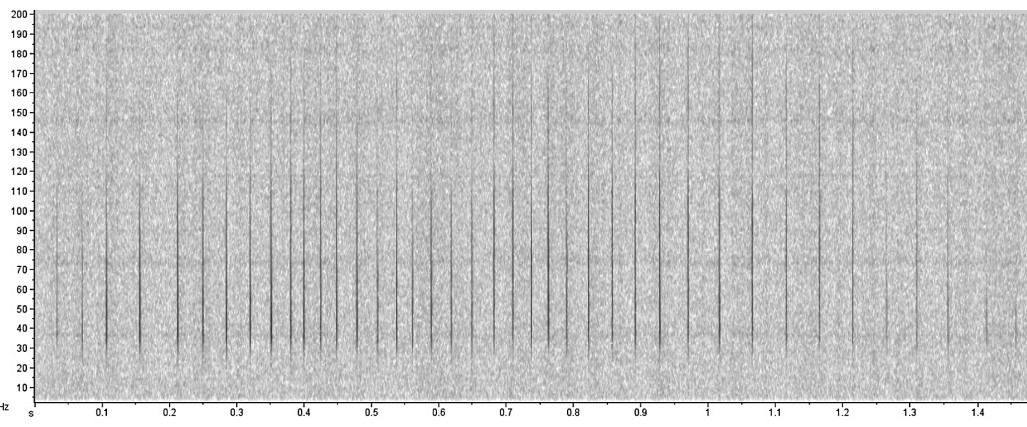

**Figure 2 Echolocation click train produced by *Inia* sp.**

and the hydrophone placed in the water approximately 2 m below the surface. The acoustic recording was done opportunistically during the sighting of an animal or group of animals. The Xingu River is the only sampled area where the distribution of *Sotalia fluviatilis* (tucuxi) overlaps with *Inia* sp. Therefore, data collection took place after a visual search to ensure that no tucuxi was observed at the time of data collection. The clear water of the Xingu River allows greater certainty that no tucuxi was close to the botos during the recordings.

Underwater sound emissions were captured using a Cetacean Research ™ C54XRS hydrophone (+3/−20 dB, −185 dB re: 1V/µPa, sampling rate of 400 kHz) of passive capture mobile. The captured sounds were transferred to a Daq/3000 Series digitizer card, and the files were recorded in .bin format and later converted to .wav (frequency response of 200kHz/24bits).

## Data analysis

The acoustic analyses were performed using the Raven Pro software 1.5 (Hamming window of 256 points of FFT with 50% overlap, Cornell Laboratory of Ornithology, New York) and MatLab R2014A (Mathworks, Natick, MA, USA). Recordings with interference and loud noise were discarded, to prevent misclassification. The low frequency noise caused by water flow was filtered with a cut-off at 5kHz and the click trains were detected through visual analysis in Raven Pro software. Then, meta-data files were created containing just one click train (Fig. 2). Next, we used a custom routine in MatLab R2014A to compute the following parameters: peak frequency, signal bandwidths (3 dB and 10 dB) and inter-click intervals (ICI). Firstly, the custom routine plots the waveform for the user to choose a threshold above which the eligible clicks have their spectrum and peak frequency calculated. Then, it computes the higher and lower 3 dB and 10 dB power points for the final calculation of the bandwidths. The user visually evaluates and chooses a different threshold for each click train. These parameters are the most used in the literature to characterize and distinguish the clicks. Multiple overlapped click trains due to the simultaneous vocalization by more than one animal were only used for frequency analyses and not for the ICI computation.

Statistical analysis was conducted in R (*R Core Team, 2015*). Firstly, the descriptive statistics were calculated for all parameters including maximum and minimum values, mean, standard deviation, median and interquartile range. Then, the Kruskal–Wallis test was applied to check if there is a difference among the 'species' and subspecies for all parameters analyzed. Subsequently, a Dunn–Bonferroni post hoc method following the Kruskal–Wallis test was also performed to discriminate the lineages (i.e. analyze each combination pair to verify the differences between them) only for the frequency parameters of the clicks.

Then, Random Forest models were created to classify the lineages according to their echolocation clicks (packages 'randomForest' and 'pROC') (version 3.4.3, *R Core Team, 2015*). Random Forest models are a series of unpruned classification trees, with 500 bootstrap samples taken from the original data set. Data not selected to build a tree were referred to as "out-of-bag" (OOB) and were used to validate classification accuracy of the forest, estimating the error rate (*Brieman, 2001*; *Liaw & Wiener, 2002*). Next, the importance of each variable (peak frequency, 3 dB and 10 dB bandwidths) was tested with the mean decrease accuracy and Gini variable importance measure. This metric is based on a weighted mean of the improvement of individual trees based on the inclusion of each variable as a predictor. We used 80% of the data for training and 20% were for testing. Finally, Receiver Operating Characteristic (ROC) curves were created in order to verify the classifying efficiency of the model by the area under the curve (AUC). We determined model performance as acceptable when AUC $\geq$ 0.7 (*Swets, 2013*).

In a first step of the classification analysis, the Random Forest model was used to classify only the lineages *I. araguaiaensis, I. g. geoffrensis* and *I. g. humboldtiana*, because the 'species' *I. boliviensis* could not be visually distinguished from *I. g. geoffrensis* and hybrids in the Madeira River. In a second step, we ran a k-means clustering analysis (packages 'factoextra', 'cluster' and 'tidyverse') only for the sampled animals in Madeira River (*I. boliviensis*, *I. g. geoffrensis* and hybrids) in order to verify if the clicks could be grouped into clusters. This method is commonly used to automatically partition a data set into k groups. It proceeds by selecting k initial cluster centers and then iteratively refining them (*Wagstaff et al., 2001*). We used the silhouette method to establish the optimal number of clusters within Madeira River. We applied the Hubert index and D index as methods to determine the best number of clusters, through the "NbClust" function (using: method = "kmeans") (package NbClust) (*Charrad et al., 2014*). Then, we used the Random Forest model again, but herein considering the k clusters and adding *I. g. geoffrensis* from the Xingu River to the analysis, in order to compare each cluster to this species, as it is also presented at the Madeira River together with *I. boliviensis* and hybrids.

Table 1 summarizes the data analyzed, with the number of click trains and clicks, the mean number of animals during the recordings, the sampling effort and the minutes analyzed in each river.

## RESULTS

*Inia araguaiaensis* clicks showed the highest peak frequency value (mean = 49.0 ± 12.0 kHz) and *I. g. humboldtiana* the smallest (mean = 43.9 ± 7.7 kHz). Both 10 dB and

**Table 1 Overview of data used in the analysis, including the mean number of animals in each sampled river.**

| River | Effort (days) | Minutes analyzed | Mean number of animals | Number of click trains | Number of clicks | Mean water depth (m) | Water type |
|-------|------|------|------|------|------|------|------|
| Araguaia | 6 | 34 | 3.7 | 41 | 1,637 | 5 | Black |
| Xingu | 5 | 28 | 4.3 | 53 | 779 | 7.5 | Clear |
| Guaviare | 6 | 6 | 3 | 24 | 1,636 | 12 | White |
| Madeira | 4 | 18 | 4.4 | 40 | 1,799 | 3 | White |
| Total | 21 | 86 | 3.8 | 158 | 5,851 | – | – |

**Table 2 Descriptive statistics of echolocation clicks of the species of genus *Inia*.** Individuals from the Madeira River (*I. boliviensis, I. g. geoffrensis* and hybrids) are represented as *Inia* spp. The mean, standard deviation, maximum and minimum values are represented for the interclick interval (ICI), 10 dB bandwidth (10 db BW), 3 dB bandwidth (3 dB BW) and peak frequency (Fp).

| Species/subspecies | Value | ICI (ms) | 10 dB BW (kHz) | 3 dB BW (kHz) | Fp (kHz) |
|-------|-------|------|------|------|------|
| *I. araguaiaensis* | mean ± sd | 39.6 ± 30.9 | 74 ± 27.6 | 32.2 ± 17.8 | 49 ± 12.1 |
| | max–min | 228.3–2.0 | 354.9-11.6 | 84.9–7.2 | 106–32.7 |
| | median | 30.7 | 76.8 | 29.4 | 46.2 |
| | interquartile range | 23.1–43.9 | 55.5–90.6 | 18.5–40.1 | 23.1–43.9 |
| *I. g. geoffrensis* | mean ± sd | 68.9 ± 35.5 | 65.5 ± 28.8 | 24.3 ± 14.8 | 45.5 ± 9.3 |
| | max–min | 202.1–10.2 | 346.2–11.1 | 81.6–7.1 | 100.5–10.6 |
| | median | 57.6 | 63.6 | 20.9 | 42.7 |
| | interquartile range | 44–84.6 | 41.5–85.1 | 12.6–30.8 | 38.6–50.4 |
| *I. g. humboldtiana* | mean ± sd | 13.8 ± 7.4 | 72.7 ± 23.6 | 28.2 ± 12.7 | 44 ± 7.3 |
| | max-min | 96.9–2.5 | 370.8–24.7 | 77.1–11 | 97.6–24.4 |
| | median | 12.4 | 72.6 | 23.6 | 41.2 |
| | interquartile range | 8.6–16.9 | 55.2–85 | 21.1–30.5 | 39.6–45.6 |
| *Inia* spp. | mean ± sd | 33.9 ± 28.4 | 77.6 ± 28.9 | 33.8 ± 20.1 | 45.5 ± 12.4 |
| | max–min | 208.6–1 | 345.4–10.6 | 84.5–6.4 | 103.1–31.0 |
| | median | 24.4 | 81.7 | 27.1 | 42.1 |
| | interquartile range | 16.9–38.6 | 53.6–100 | 18.8–46.2 | 38–47.9 |

3 dB bandwidths were higher for *I. boliviensis, I. g. geoffrensis* and hybrids from the Madeira River (mean = 77.6 ± 28.9 kHz and 33.8 ± 20.1 kHz, respectively) and lower for *I. g. geoffrensis* on Xingu River (mean = 65.5 ± 28.8 kHz and 24.3 ± 14.8 kHz, respectively). *I. g. geoffrensis* showed the highest ICI value (mean = 68.9 ± 35.5 ms) and *I. g. humboldtiana* the smallest (mean = 13.8 ± 7.4 ms) (Table 2).

According to the Kruskal–Wallis test, there was a significant statistical difference among lineages for all parameters analyzed ($p$-value < 0.05). According to the post hoc test, the 10 dB bandwidth did not show significant differences for *I. araguanaensis* and *I. g. humboldtiana*, the 3 dB bandwidth did not show significant differences for *I. araguaiaensis* and *Inia* spp. (*I. boliviensis, I. g. geoffrensis* and hybrids), and the peak frequency was not significantly different comparing *I. g. humboldtiana* with *Inia* spp. and
**Table 3 Discrimination of echolocation clicks parameters between 'species' and subspecies of genus *Inia* by Dunn-Bonferroni post hoc test.** The analyzed parameters were: peak frequency (Fp), 10 dB bandwidth (10 dB BW) and 3 dB bandwidth (3 dB BW). *Inia* spp. represents the Madeira River population (*I. boliviensis, I. g. geoffrensis* and hybrids). *p*-values in bold show significant differences.

| Parameter | *I. araguaiaensis* X *I. g. humboldtiana* | *I. araguaiaensis* X *Inia* spp. | *I. araguaiaensis* X *I. g. geoffrensis* |
|---|---|---|---|
| Fp (kHz) | **<0.001** | **<0.001** | **<0.001** |
| 10 dB BW (kHz) | 0.08 | **<0.001** | **<0.001** |
| 3 dB BW (kHz) | **<0.001** | 1 | **<0.001** |

| Parameter | *I. g. humboldtiana* X *Inia* spp. | *I. g. humboldtiana* X *I. g. geoffrensis* | *I. g. geoffrensis* X *Inia* spp. |
|---|---|---|---|
| Fp (kHz) | 1 | 0.14 | **<0.005** |
| 10 dB BW (kHz) | **<0.001** | **<0.001** | **<0.001** |
| 3 dB BW (kHz) | **<0.001** | **<0.001** | **<0.001** |

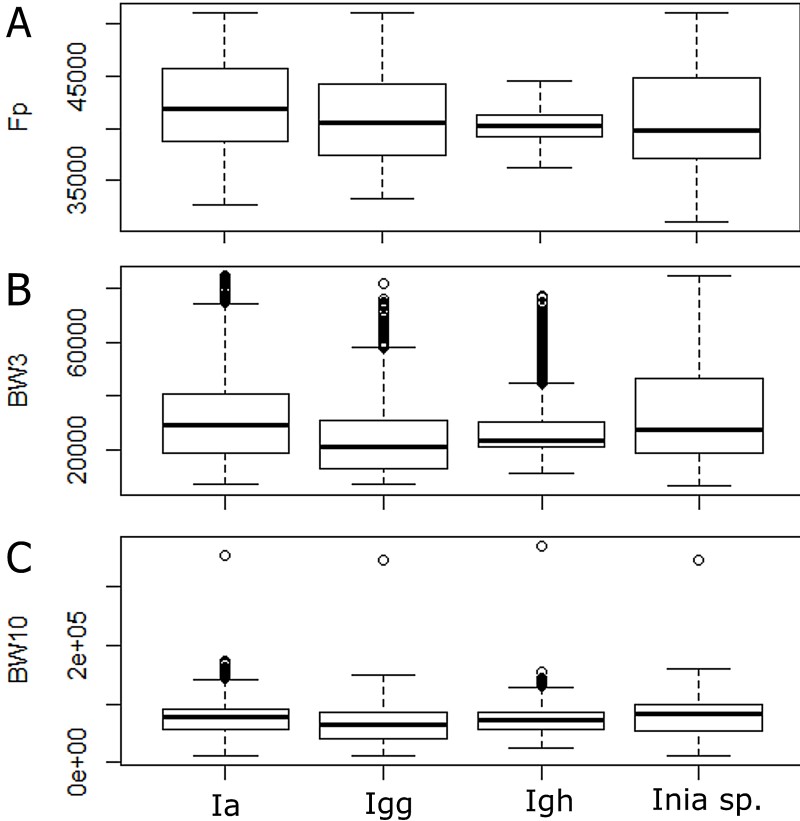

**Figure 3 Differences among echolocation clicks of *Inia* lineages.** Ia: *Inia araguaiaensis;* Igg: *I. geoffrensis geoffrensis;* Igh: *I. g. humboldtiana;* Inia sp.: *I. boliviensis, I. g. geoffrensis* and hydrids. (A) Fp: peak frequency; (B) BW3: 3 dB bandwidth; (C) BW10: 10 dB bandwidth.

*I. g. geoffrensis*. In Table 3, it is possible to see all the other pairs compared that had a significant difference in each analyzed parameter (numbers in bold). The boxplot of each analyzed parameter of the clicks shows the differences among the lineages (Fig. 3).

**Table 4 Confusion matrix of the Random Forest models.** It shows the correct classification of the echolocation clicks of genus *Inia*, as well as the misclassification. Values are shown in percentages.

| Species/subspecies | Ia | Igg | Igh | |
|---|---|---|---|---|
| Ia | 67 | 18 | 15 | |
| Igg | 23 | 68 | 9 | |
| Igh | 16 | 1 | 74 | |
| Accuracy = 70% | | | | |

Ia: *Inia araguaiaensis*; Igg: *I. g. geoffrensis*; Igh: *I. g. humboldtiana*

| Species/cluster | Igg | Ispp1 | Ispp2 | Ispp3 |
|---|---|---|---|---|
| Igg | 64 | 27 | 8 | 1 |
| Ispp1 | 20 | 79 | 1 | 0 |
| Ispp2 | 19 | 1 | 80 | 0 |
| Ispp3 | 13 | 0 | 2 | 85 |
| Accuracy = 76% | | | | |

Igg: *I. g. geoffrensis* from Xingu River; Ispp1, Ispp2 and Ispp3: clusters from k-means analysis with individuals from Madeira River (*I. boliviensis, I. g. geoffrensis* and hybrids)

Random Forest model showed low misclassification among the lineages analyzed—minimum of 9% between *I. g. geoffrensis* and *I. g. humboldtiana* and maximum of 23% between *I. g. geoffrensis* and *I. araguaiaensis* (Table 4)—and a clear separation of *I. araguaiaensis, I. g. geoffrensis* and *I. g. humboldtiana* by their echolocation clicks (Fig. 4). The general accuracy of the model was of 70%, and the balanced accuracy for *I. araguaiaensis* was of 75%, for *I. g. geoffrensis* was 69% and for *I. g. humboldtiana* was 81%. The parameters that most contributed to the model were peak frequency and 3 dB bandwidth. Random Forest classifier showed high goodness of fit with area under the curves of 0.897 for *I. g. humboldtiana*, 0.837 for *I. araguaiaensis* and 0.793 for *I. g. geoffrensis*.

Regarding data analysis of the Madeira River (*Inia* spp.), we found three clusters as an optimal number of clusters by the silhouette method (Fig. 5). This may be due to the presence of three groups of animals in the area where we collected data–*I. boliviensis, I. g. geoffrensis* and hybrids. We termed the clusters as Ispp1, Ispp2 and Ispp3, as we could not certainly assign them to *I. g. geoffrensis, I. boliviensis* or hybrids. The Random Forest analysis, performed with *I. g. geoffrensis* (Igg) from the Xingu River together with the clusters, showed that Igg had 64% of correct classifications and 27% of misclassification with Ispp1, which classified correctly in 79% of the data. Ispp2 had 80% of correct classifications and 19% of error with Igg. Ispp3 had 85% of correct classification and 13% of error with Igg (Table 4). The general accuracy of the model was 76% and the balanced accuracies were 69% for Igg, 85% for for Ispp1, 92% for Ispp2 and 97% for Ispp3. The classification tree is represented in Fig. 6. The parameters that most contributed to the model were peak frequency and 10 dB bandwidth. The ROC curves of this classification showed the goodness of fit of the model with areas greater than 0.809 (Fig. 6).

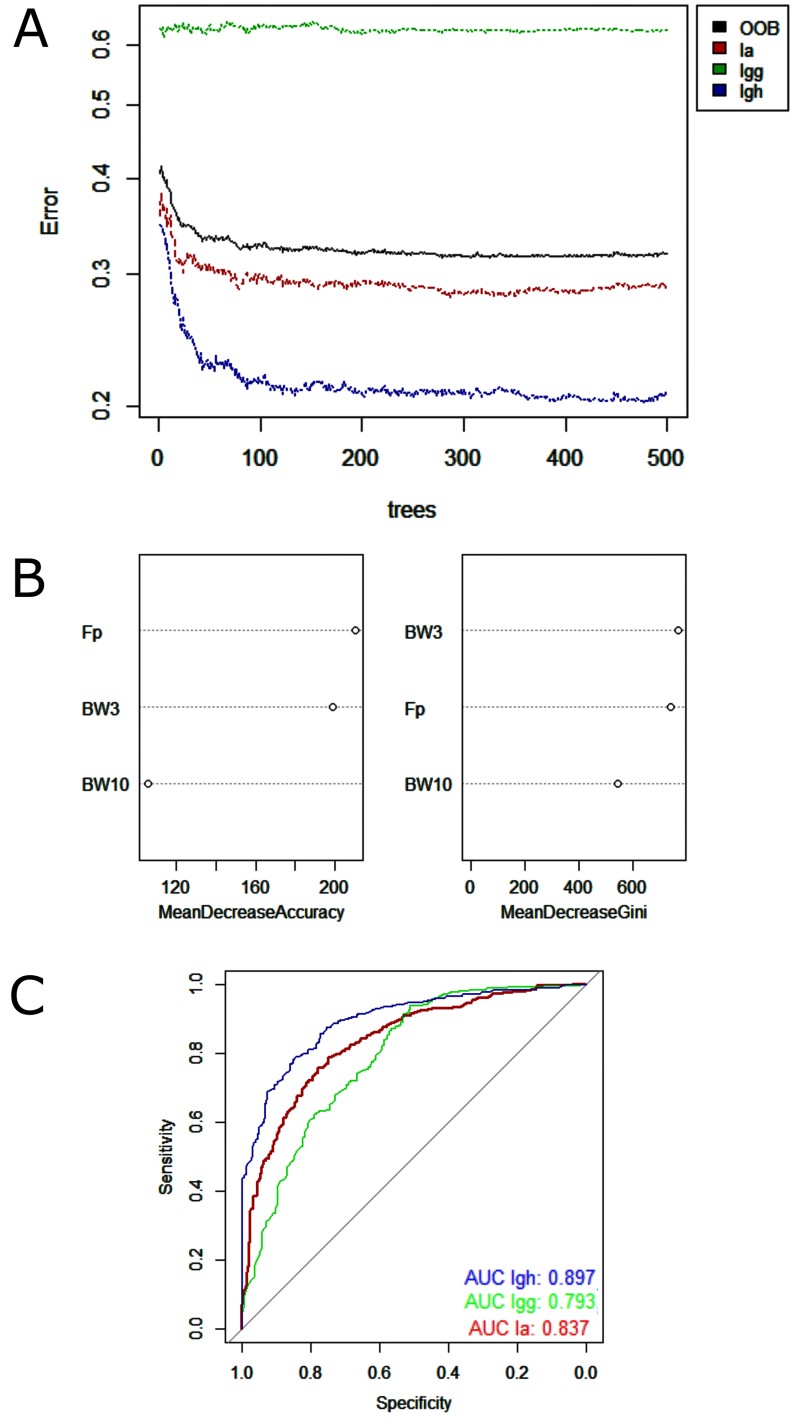

**Figure 4 Outputs of the Random Forest models for the classification of the 'species' and subspecies of genus *Inia* according to their echolocation clicks.** Ia: *Inia araguaiaensis*; Igg: *I. geoffrensis geoffrensis*; Igh: *I. g. humboldtiana*. (A) Learning curves of the decision trees with an out-of-bag estimator (OOB) of 31.84%; (B) Mean decrease accuracy and Gini variable importance measure showing the importance of each analyzed vocalization parameter (Fp: peak frequency; BW3: 3 dB bandwidth BW3; BW10: 10 dB bandwidth) for the model; (C) Receiver Operating Characteristic (ROC) curves: each curve represents the sorting of the efficiency of the model for the 'species' and subspecies and the area under the curve (AUC) is the indicator of the goodness of fit.

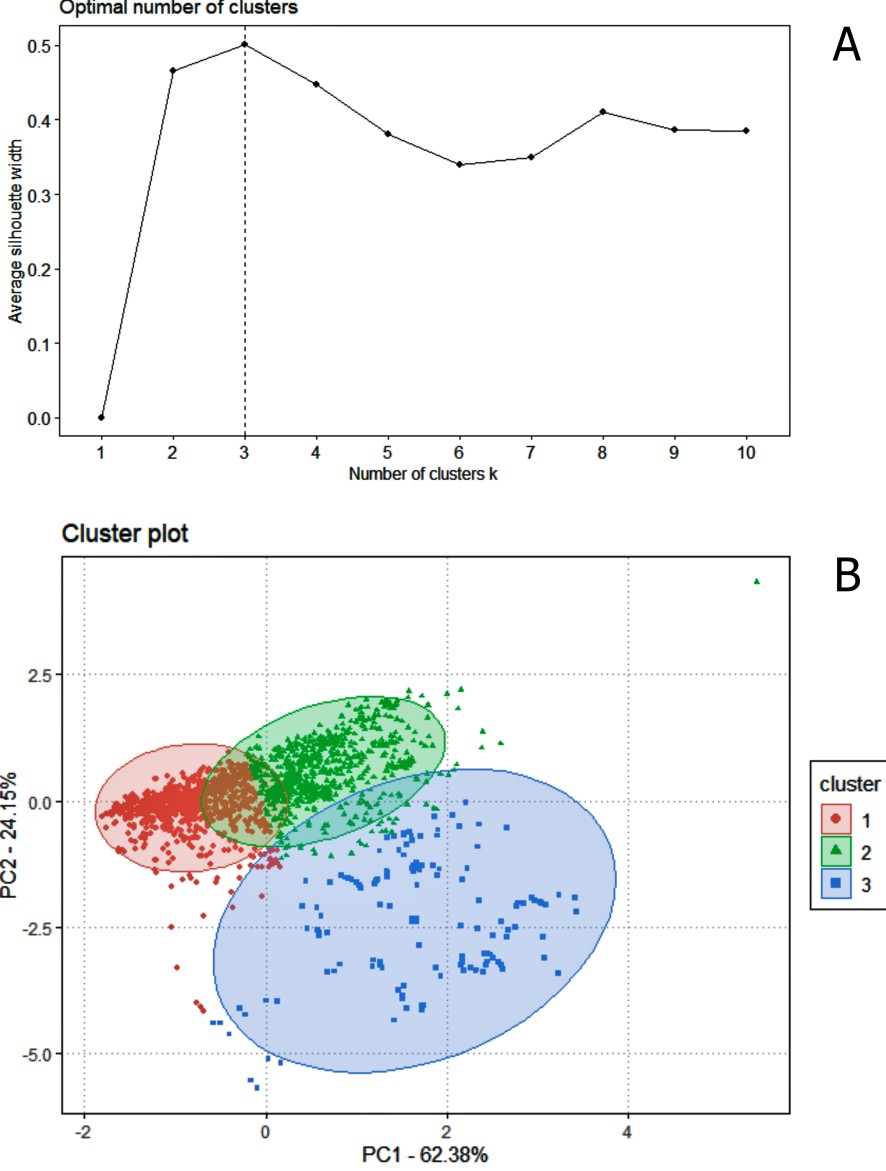

**Figure 5 The k-means clustering analysis for the animals of the Madeira River (*I. boliviensis*, *I. g. geoffrensis* and hybrids).** (A) Shows the silhouette analysis with the optimal number of clusters (three) and (B) the cluster plot of the echolocation clicks of the animals.

## DISCUSSION

### Characterization of the echolocation clicks

In the past few years, a greater effort has been made to understand the acoustic behavior of Amazon river dolphins (e.g. *Caldwell, Caldwell & Evans, 1966*; *Kamminga et al., 1993*; *Ding, Würsig & Leatherwood, 2001*; *Podos, Da Silva & Rossi-Santos, 2002*; *Diazgranados & Trujillo, 2002*; *May-Collado & Wartzok, 2007*; *Yamamoto et al., 2015*; *Ladegaard et al., 2015*, *2017*; *Amorim et al., 2016*; *Melo-Santos et al., 2019*, *2020*), but there are still no studies on the echolocation clicks of *I. boliviensis* and *I. araguaiaensis*. Most of the studies on boto's clicks (e.g. *Ladegaard et al., 2015*, *2017*; *Yamamoto et al., 2015*) describe only

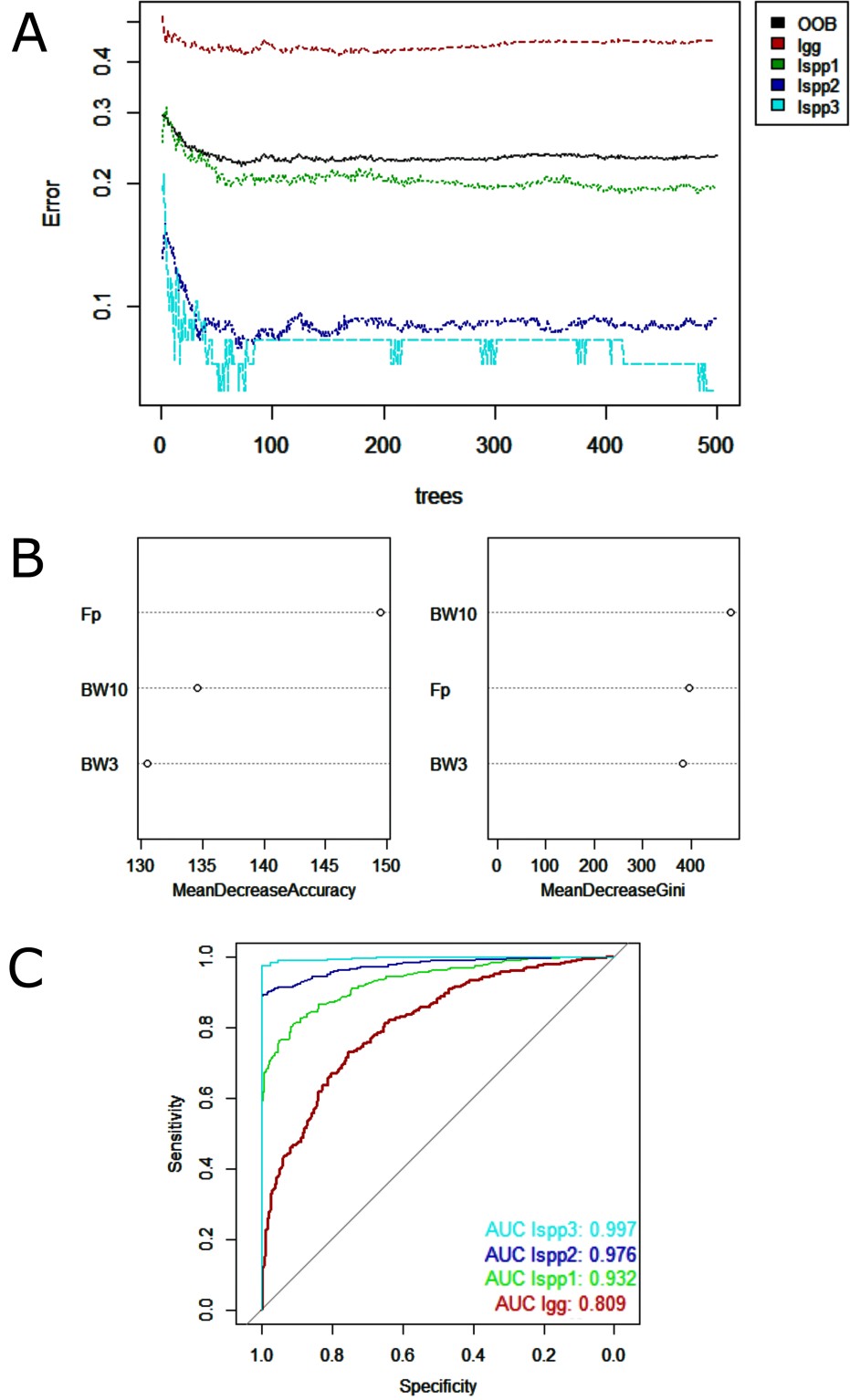

**Figure 6 Outputs of the Random Forest models for the classification of the Madeira River individuals (*I. boliviensis, I. g. geoffrensis* and hybrids) and *I. g. geoffrensis* (Ia) from Xingu River by their echolocation clicks.** Ispp1, Ispp2 and Ispp3 represents the three clusters of Madeira River individuals. (A) Learning curves of the decision trees with an out-of-bag estimator (OOB) of 23.41%;

**Figure 6** (continued)
(B) Mean decrease accuracy and Gini variable importance measure showing the importance of each analyzed vocalization parameter (Fp: peak frequency; BW3: 3 dB bandwidth BW3; BW10: 10 dB bandwidth) for the model; (C) Receiver Operating Characteristic (ROC) curves: each curve represents the sorting of the efficiency of the model for the 'species' and subspecies and the area under the curve (AUC) is the indicator of the goodness of fit.       

clicks recorded on the animal's body axis, unlike the present study, where we analyzed both on and off-axis clicks. On-axis clicks may not accurately represent the complete set of clicks that are acquired during passive acoustic monitoring of odontocetes (*Soldevilla et al., 2008*). *Au, Floyd & Haun (1978)* established that off-axis click durations are longer, usually due to multiple paths of the initial click pulse, and suggested that the multiple paths are due to reflections within the head, the external environment, or a combination of the two. *Amorim et al. (2019)* discriminated eight delphinid species by their off-axis echolocation clicks and found that these pulsed sounds are better when comparing to the tonal sounds in discriminating species.

The farther the click is recorded off the animals' axis, both horizontally and vertically, the lower frequency will be the strongest peak in the spectra (*Au, 1980*). Here, we found a mean peak frequency of 45.5 kHz for *I. geoffrensis'* clicks, as well as *Kamminga & Wiersman (1981)*, that found *I. geoffrensis'* echolocation clicks at 40–80 kHz. However, our results showed peak frequency values dropping almost by half comparing to 96 kHz found by *Ladegaard et al. (2017)* and 82 kHz by *Yamamoto et al. (2015)*. The river dolphin *Sotalia fluviatilis* (tucuxi), which overlaps its area of occurrence with *I. geoffrensis*, produces clicks with a peak frequency around 80–90 kHz (*Kamminga et al., 1993*; *May-Collado & Wartzok, 2010*). The presence of this animal may be a bias in the study of the bioacoustics of the boto. In the current study, tucuxi's area of occurrence overlaps with *I. geoffrensis* only in the Xingu River, but care was taken to visually detect the presence of animals at the time of recording. In addition, different data collection equipment can also influence the result. Therefore, we verified the need for standardization in studies and further investigations among populations of river dolphins throughout the Amazon.

In relation to other species of river dolphins around the world, the *Lipotes vexillifer* (baiji) produces clicks with peak frequency between 50 and 100 kHz (*Akamatsu et al., 1998*), the *Pontoporia blainvillei* (franciscana dolphin) produces high frequency clicks (+/− 139 kHz, *Melcón, Failla & Iñíguez, 2012*), and the *Platanista gangetica gangetica* (Indus river dolphin) has an average peak frequency of 58.8 ± 6.8 kHz (*Jensen et al., 2013*). Both the Indus river dolphin and the boto face challenges to locate food and move around in an acoustic habitat with high levels of reverberation and attenuation. Vocalizing in lower frequencies can guarantee that the acoustic information is transmitted reliably, increasing the active space of the signal under conditions of greater attenuation and dispersion (*Hamilton et al., 2001*). Previous studies partly support this hypothesis that the peak frequency of Amazon river dolphin clicks that inhabited flooded forests was slightly lower compared to the *Sotalia fluviatilis's* clicks, species that does not go into the flooded forests of the Amazon (*Yamamoto et al., 2015*).

## Differences in the species and subspecies of the genus *Inia*

All the parameters of the clicks analyzed in this work (ICI, peak frequency, 3 dB and 10 dB bandwidth) were significantly different between lineages of the genus *Inia*. However, we only use the parameters in the frequency domain, since the ICI depends on the behavioral context of the animal (*Madsen et al., 2005*; *Baumann-Pickering et al., 2010*). The lineages of genus *Inia* are morphologically distinguishable through cranial measurements and number of teeth (*Hamilton et al., 2001*; *Hrbek et al., 2014*). The skull characteristic can influence the sound production path (*Walker et al., 1986*). These dolphins are not visually distinguishable, and the ability to distinguish them acoustically could offer a view of the differences in the biology of each lineage.

The Araguaian boto, a species not yet confirmed by the Committee on Taxonomy of the Society for Marine Mammalogy, had all the click frequency parameters significantly different from *I. g. geoffrensis*. The misclassification of sounds between the two lineages according to the Random Forest analysis, was also low (18% and 23%), showing a potentially useful evidence of taxonomic distinction. Due to the aforementioned factors of association of the animal's skull shape in relation to the click frequency characteristics (*Lilly & Miller, 1961*; *Norris, 1968*, *1975*; *Norris et al., 1971*), in addition to studies that classify species through the echolocation clicks (e.g. *Baumann-Pickering et al., 2010*; *Amorim et al., 2019*), our results may be another evidence that there are differences between both lineages, reinforcing the classification of *I. araguaiaensis* as a distinct species from *I. geoffrensis.*

There was a greater difference in frequency between different 'species' than between individuals of the same species, since the subspecies *I. g. geoffrensis* and *I. g. humboldtiana* did not show significant differences in peak frequency of the clicks according to the post hoc test. However, the 3 dB and 10 dB bandwidths were significantly different between subspecies, showing a discriminating aspect between their echolocation clicks. Additionally, *I. g. humboldtiana* had the higher rate of correct classification (74%) according to the Random Forest model. Efforts are being made to obtain more information on the possible classification of *I. g. humboldtina* as a new species, or at least as a separate evolutionary unit from *I. g. geoffrensis* (*Trujillo et al., 2004*; F. Trujillo, 2020, personal communication). Our results can assist in the classification of these dolphins.

The clusters generated by k-mean analysis showed the possible existence of three distinct groups of sounds collected in the Madeira River. This data was collected in a hybrid zone of *I. g. geoffrensis* and *I. boliviensis*. It is possible that these clusters are associated with each of these animals. The formation of an overlap zone between *I. boliviensis* and *I. g. geoffrensis* is natural and has not occurred recently (*Gravena et al., 2015*; J. Farias & G. Melo-Santos, 2020, personal communication), although it was forced by the construction of the dams (Jirau and Santo Antônio hydroelectric plants). The hybrid population of *Inia* sp. would be biologically distinct from the species that originated it since it is expected that hybrid animals will also develop specific characteristics, which was confirmed through our cluster results, supporting the possible existence of three distinct acoustic groups in this area. These findings support the
hybridization hypothesis. In order to have a greater degree of certainty about the animals of the Madeira River, it is necessary to record in a region where only *I. boliviensis* is present, i.e. above the Jirau hydroelectric plant near Abunã, where the occurrence of *I. geoffrensis* is already ruled out (*Gravena et al., 2015*).

Automatic click classifiers have not yet been tested for river dolphins in South America. We present evidence that the clicks of *Inia* sp. have specific and promising characteristics to be automatically detected for the use in passive acoustic monitoring. However, even though we have potentially useful evidence of taxonomic distinction, the misclassification between the lineages would substantially limit the accuracy or applicability of acoustic monitoring. In addition, as lineages of *Inia* are geographically separated, the key tasks for passive acoustic monitoring of river dolphins from the Amazon will be low error rates in achieving distinction from *Sotalia fluviatilis* and from non-cetacean noise sources. Therefore, our results are preliminary and further investigation on a broader dataset is necessary.

## Conservation

*Inia geoffrensis* is classified as "endangered" on the IUCN Red List (*da Silva & Martin, 2018*). There is still no conservation status for *I. boliviensis* and *I. araguaiaensis* due to the lack of knowledge on distribution range, population estimates, genetics, and threats for these species. *Gomez-Salazar et al. (2012)* suggest independent status for geographically distinct populations of the Bolivian boto, separated by different hydrographic basins (the upper Madeira River in Brazil, and the Itenez-Mamoré river basin in Bolivia). The Araguaian boto (*Inia araguaiaensis*) appears as the most distinct from its counterparts, with low levels of genetic diversity, in addition to the restricted distribution in a highly fragmented riverine-scape, and possibly presenting low population numbers compared to the Amazon boto (*Hrbek et al., 2014*; *Paschoalini et al., 2020*). Such evidences, summed with the lack of dedicated studies to these lineages, are quite concerning.

All lineages of Amazon river dolphins are threatened by human activities, i.e. hydroelectric constructions and conflicts with fisherman (*Iriarte & Marmontel, 2013*; *Pavanato et al., 2016*; *Paschoalini et al., 2020*). For the correct evaluation of the impacts of such threats on the 'species' or populations, so as the proper formulation of conservationist actions and environmental policies, it is advisable to assign the conservation status of the lineages based on the characters described in the literature, its distribution and population numbers, and also the findings of the present study. If a species is included in the IUCN red list, for example, it will be prioritized on conservation studies. The results of the present study have shown to be useful as a tool for the differentiation among lineages of genus *Inia*, contributing to the few morphological differences. Classifying *I. araguaiaensis* separately from *I. geoffrensis*, specially, is substantially important due to the pressure of human activities in the Tocantins-Araguaia river basin (mainly dams). Once classified, further studies on distribution and population estimation may provide greater knowledge about its conservation status, and thus provide protective measures for the new species.

## CONCLUSION

Amazon river dolphins (*Inia* spp.) have shown species-specific acoustics properties in their clicks. Their echolocation clicks had significant differences between lineages, thus acoustics approaches can be an effective tool to differentiate *Inia* species. This study presents more evidence of differences between the newly described *I. araguaiaensis* from *I. geoffrensis*. Our results may assist in the passive acoustic monitoring of dolphins and possibly improve efforts and knowledge for *I. g. humboldtiana*. However further studies are needed to analyze *I. boliviensis* separately, and to investigate inter- and intra-species variations based on their acoustic parameters.

## ACKNOWLEDGEMENTS

We are thankful to Santo Antônio Energia (Fauna Conservation Program in Madeira River); to Sete Soluções e Tecnologia Ambiental Ltda, in the person of Eduardo Sábato; to Danielle Lima (research coordinator of the Aquatic Mammal Monitoring Program in Madeira River); to Norte Energia (Aquatic Mammal Monitoring Program in Xingu River); to LEME Engenharia Ltda; to Omacha Fundation, in the person of Fernando Trujillo; to Joice Farias and Gabriel Melo-Santos for the personal comments; to Adailton Glória (Cantão State Park manager); to Samara Bezerra and Bruna Pagliani for the help in the fieldwork; and to Anne Landine for the conceptualization of the map.

### Funding

This work was supported by the CAPES Foundation from the Brazilian Ministry of Education provided master's and doctoral scholarships (No. 1092653 and 1571839 respectively), the Universidade Federal de Juiz de Fora (UFJF) and the Cetacean Society International. There was no additional external funding received for this study. The funders had no role in study design, data collection and analysis, decision to publish, or preparation of the manuscript.

### Grant Disclosures

The following grant information was disclosed by the authors:
CAPES Foundation: 1092653 and 1571839.
Universidade Federal de Juiz de Fora (UFJF).
Cetacean Society International.

### Competing Interests

The authors declare that they have no competing interests.

### Author Contributions

- Jéssica F. Melo conceived and designed the experiments, performed the experiments, analyzed the data, prepared figures and/or tables, authored or reviewed drafts of the paper, and approved the final draft.

- Thiago O. S. Amorim analyzed the data, prepared figures and/or tables, authored or reviewed drafts of the paper, and approved the final draft.
- Mariana Paschoalini conceived and designed the experiments, performed the experiments, authored or reviewed drafts of the paper, and approved the final draft.
- Artur Andriolo conceived and designed the experiments, authored or reviewed drafts of the paper, and approved the final draft.

### Field Study Permissions

The following information was supplied relating to field study approvals (i.e., approving body and any reference numbers):

Field experiments in the Cantão State Park were approved by the Tocantins State Government, Instituto Natureza do Tocantins - Naturantins (permit number: 1497-2017).

### Data Availability

Raw data are available in the Supplemental Files.

### Supplemental Information

Supplemental information for this article can be found online at http://dx.doi.org/10.7717/peerj.11105#supplemental-information.

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
