# Peer review of "The biosonar of the boto: evidence of differences among species of river dolphins (Inia spp.) from the Amazon"

_PeerJ, doi:10.7717/peerj.11105_

## Round 0.1 · original submission · Major Revisions

Dear Authors,

I received three reviews of your MS. All the reviewers were positive, but there are still questions that need to be addressed and suggestions to be implemented. Taking these suggestions into account, you will have a solid MS.
Job well done, and I look forward to receiving your revision shortly.

Sincerely,

Tomas Hrbek

·

Basic reporting

The paper presents a valuable set of consistently gathered high-resolution acoustic data from widely dispersed sites and all the significant taxonomic candidates.
The findings are relevant to taxonomy and to passive acoustic monitoring. The level of discrimination of taxons for these tasks is significantly different - a level of misclassification of say 10% still leaves potentially useful evidence of taxonomic distinction, but will substantially limit the accuracy or applicability of acoustic monitoring. This could be usefully discussed.
Only a small sample of raw data from one animal is provided.

Experimental design

The sampling rate or rates used need to be stated.
The custom routine used to derive values for bandwidth needs more description. Clicks in the data sample show bimodal spectra but it is not clear how this was handled.
Clicks in the data sample also show, in the waveform, that there is a frequency trend within the click that would be seen in a Wigner plot. It is possible that this might be another discriminating feature.
Looking at the waveform it does appear that the bimodality and temporal trend in frequency are likely to be related with a louder lower frequency early component and a weaker later higher frequency component. Each may represent some natural resonance in the acoustic pathway within the animal and be relevant to the task in hand.
'we analysed both on and off-axis clicks' As the earliest and latest clicks in a recorded 'train' are generally off-axis this could be used to get an more information on the beam structure of the taxonomic groups. The raw data of a single train provided does show less high frequencies in the off-axis clicks.
The choice of 3 clusters from the k-means analysis is based on the silhouette analysis which slightly favours 3 clusters over two. However the decision line this creates (Figure 4) between clusters 1 and 2 is not very plausible, while a classification into (1 and 2), (3) would be much more plausible. This is an arguable choice, but in the context of the general risk of over-classifying where the data may not be fully representative I would argue strongly for 2 clusters to be used.

Validity of the findings

The main limitation of this is clearly recognised by the authors and is the limited sample sizes, and the spread over seasons and locations. If the acoustic characteristics of the clicks are adapted by the dolphins to fit environmental conditions, including prey types and sizes, then the generalization performance may be compromised, and the OOB errors in random forest cannot reflect that variation that is outside the set of samples. However this data remains a valid contribution to the development.
The limitation recognized by the authors applies to both the identified tasks. In the case of application to passive acoustic monitoring it would be relevant to note that in many locations the key tasks will be low error rates in achieving distinction from Sotalia and from non-cetacean noise sources.

re Model performance is considered acceptable when AUC 0.7 (Swets, 2013). The acceptable level of performance depends on the task in hand. Where very low FP rates are required AUC is itself of limited value.

Additional comments

Valuable data is presented very clearly, and these notes are on minor textual issues:

Abstract
Echolocation clicks can reveal the anatomy of the vocalizing animal, enabling the distinction of species.
>> 'reflect the anatomy' might be more accurate as we do not know, at present, how to predict anatomy from the subtle acoustic features reported.
Differences in acoustical parameters of clicks are mainly related to skull morphology
>> that is a bold claim, given that the acoustic properties of the soft tissues of the acoustic pathway are as significant as the bony tissues.

56 their clicks, mainly with the increasing use of passive acoustic monitoring
>> 'mainly in response to the increasing'
78 thereby
>> and
79 79 responsible for their separation are subtle, the Araguaian and Bolivian botos are still for being recognized by
>> needs re-phrasing
87 Until then, we are not aware of studies that
>> We are not aware

100 boliviensis was sampled in the Madeira River withing
>> within

108 the upper reaches of the river, low nutrient is available, with rapid flow of sediments and sandy
>> river nutrient levels are low, there is rapid flow and sandy sediments.

115 (várzea) (Latrubesse & Sinha, 2005), being the Itamacará waterfall the limit for river dolphins’ distribution
>> 2005). The Itamacará waterfall is the upper the limit of the dolphins’ distribution

117 and one of the main tributaries of the Amazon river basin, extends to three countries
>> in the Amazon basin extending to

122 tributary of the Tocantins-Araguaia basin. It is a low depth-black-water river with 2,600 km in length
>> omit with

139 For data collection, a small outboard vessel was used for displacement and dolphins’ observation.
>> displacement?

141 in the water, 2 meters depth approximately.
>> in the water approximately 2m below the surface.

159 and the click trains were detected trough visual analysis in Raven Pro software.
>> through

166 coefficient of variation. Then, Kruskal-Wallis test was
>> Then the K...

252 frequency will be the strongest peak in the spectra (Au, 1980).
>> Au does say that, but I have measured this from Botos and Tucuxi and not found it to be uniformly true. Not published!

Reviewer 2 ·

Basic reporting

Melo et al. reported differences in the biosonar parameters between Inia spp. The authors should be more cautious when suggesting that differences in “biosonar” could differentiate species. First, because there are few individuals per lineages. Secondly, because there are different types of "calls" that could indicate from a specific type of prey to a stick floating in the river (for example turtles have a large intraspecific repertoire of "calls”). The authors do not specify if the audios were recorded in the breeding season, which leaves many doubts if the acoustic parameters are related to some type of reproductive isolation. The ontogeny of specimens also can be a problem. Did the authors know if they recorded juveniles or adults, males and females?
In addition, the authors report not differences between subspecies. If we follow the main idea of the MS that is differentiate lineages by biosonar parameters, the subspecies must disappear. This is not good for conservation, because the subspecies have their own evolutionary history. In addition, the authors report that hybrids between I. geoffrensis and boliviensis have divergences in the biosonar parameters. Based on their title (“…a tool to distinguish the species of river dolphins from the Amazon”) we have a new species? Also, I strong recommended that the authors use lineages and not “species” or “subspecies”.
Despite this, Melo et al. presented an interesting study that arise new taxonomic and ecological questions for Iniia spp.
In methodology, there are few recording specimens by lineages. I suggest a PCA or DFA (discriminant function analyses) including all the Iniia spp. and not only the lineages of Madeira. This will be supporting the evidence that there is difference between lineages in biosonar parameters. I also suggest a boxplot between lineages using the peak frequency. This will facilitate the readers note the biosonar differences between botos.
The results seem as a comparison between “acoustic” parameters of species. I suggest the authors describe the bioacoustics parameters per lineages, and include (if they have) the environmental conditions of the recording (i.e. water temperature, rain or sunny day, season of record, etc.). Peak frequency is the same than dominant frequency? The authors did not discriminate between them.
In discussion the authors are refuting their own work (322-339) when they say that the variation in the acoustic parameters can be caused by a multitude of variables, from ecological to evolutionary. Based on the biological species concept the “calls” must prevent the breeding between lineages. I missed discussing more relevant topics from the point of view of discrimination between species of mammals using the biosonar parameters. The authors can discuss the relevance of reproductive isolation and the use of acoustic parameters in identifying the proper lineage.
All the others suggestions are in the annotated MS.

Since English is not my first language, I ask the authors be careful by the paragraphs written by me.

Experimental design

Work with river dolphin are very difficult. The collected data is extraordinary and new. However, there is a few specimens per linages and absent of data as ontogeny of recording specimens.

Validity of the findings

This is an important contribution for evolutionary ecology in the genus Iniia

Additional comments

The MS needs a new focus. Describe and compare the biosonar parameters of Iniia lineages, made a PCA or DFA and show the differences. In discussion, explore the importance of these divergences in an evolutionary and species delimitation context.

Annotated reviews are not available for download in order to protect the identity of reviewers who chose to remain anonymous.

Reviewer 3 ·

Basic reporting

The paper is clear and professionally structured. I made a few comments on the manuscript to improve the English.

Experimental design

The range in frequency response of the hydrophone is pretty high (+3/-20 dB), did you correct these values with the transfer function? Would it be possible that the lower frequencies reported in this paper compared to previous papers are also related to the lower hydrophone sensitivity beyond 44 kHz when you loose flat response of your hydrophone? I think this aspect should be clarified.
Also, I assume that your data is not normally distributed so why do you report mean values instead of medians? I think medians will describe better your data, specially for comparison with other studies and species.

Validity of the findings

I have concerns related to how the non-flat frequency response of the hydrophone beyond 44 kHz can affect the frequency parameters reported in this paper, specially when comparing with other studies. This aspect needs further clarification in the paper.

Additional comments

I like the statistical approach of this paper and think results showed in this paper are relevant as a first step for further studies applying passive acoustic monitoring on these endangered Amazon river dolphins.

Annotated reviews are not available for download in order to protect the identity of reviewers who chose to remain anonymous.

---

## Round 0.2 · Minor Revisions

Dear Authors,

I received two reviews of your revised MS, and I have reviewed your MS myself. First, you have done an admirable job in addressing reviewer comments and implementing relevant suggestions/corrections. I am happy this versions, but there are few minor issues that I would like to see addressed, and that should be quick to implement.

1. Check your references (species names in italics, journal/report names every word capitalized, only the first word of article names is capitalized, etc.)

2. Figures 3 to 6 are of very poor quality. Given your proficiency in R, please use ggplot2 or similar to produce high-quality raster format figures.

3. As supporting material, please provide the R scripts that you used in your analyses (principally implementation of the Random Forest models and the k-means clustering analysis).

4. The “peerj-52401-results_boto.xlsx” supplemental material is just a text file pasted into an Excel with each row occupying one cell. Report these results in either a Word file, or better yet just as the text file that it is.

Few specific observations:
L143 – “Jaci-Paraná” not “Jaci-Paranã”
L151 – “dolphin observations” not “dolphins’ observation” - this is a common mistake, you are making observations of dolphins (dolphin observations). “dolphins’ observation” means that the observations belong to the dolphins.
L152 – “the engine of the vessel” not “the vessel's engine”
L276 – “bioacoustics of the boto” not “boto's bioacoustics”
L337 - “lineages of Inia” or “Inia lineages” not “Inia's lineages”

Congratulations on a job well done, and I look forward to receiving your revision shortly.

Sincerely,

Tomas Hrbek

Reviewer 2 ·

Basic reporting

The authors improve considerably the manuscript. The structure of the MS are well done. There are some puntual suggestions made by me in the introduction and methodology.

My main concern is in the figures. They seem to be little worked for such an important publication on the ecology and evolution of the river dolphins.

I suggest improving the figures (see my comments in the MS). The authors could edit them to improve the quality of the images and make them more impressive.

Experimental design

well done

Validity of the findings

Melo et al. presented an interesting study that arise new taxonomic and ecological questions for Inia spp.

Additional comments

Well done, the MS was improve. however, I strongly recommend improve the figures (see my comments in the MS). This work is impressive and the figures have to be too. The authors could place images of the environments where the samples were taken, of the species of river dolphins in relation to their recordings, etc. be more imaginative.

Annotated reviews are not available for download in order to protect the identity of reviewers who chose to remain anonymous.

Reviewer 3 ·

Basic reporting

The paper is clear and professionally structured.

Experimental design

The authors addressed well previous comments from reviewers.

Validity of the findings

This is an important contribution for evolutionary ecology in the genus Inia and as a first step for further studies applying passive acoustic monitoring on these endangered Amazon river dolphins, but standardization of methogology applied in future studies would be key.

Additional comments

Valuable preliminary data is presented, hopefully continuation of this studies will improve understanding and conservation status of Amazon river dolphins.

---

## Round 0.3 · accepted · Accept

Dear authors,
Thank you for your revision. I am now happy to accept your paper for publication.

Congratulations on a job well done.
Tomas Hrbek